# The Role of *SOCS3* in Regulating Meat Quality in Jinhua Pigs

**DOI:** 10.3390/ijms241310593

**Published:** 2023-06-24

**Authors:** Fen Wu, Zitao Chen, Zhenyang Zhang, Zhen Wang, Zhe Zhang, Qishan Wang, Yuchun Pan

**Affiliations:** 1College of Animal Sciences, Zhejiang University, Hangzhou 310058, China; 2Key Laboratory of Livestock and Poultry Resources Evaluation and Utilization, Ministry of Agriculture and Rural Affairs, Hangzhou 310058, China

**Keywords:** meat quality, cell proliferation, glucose uptake, multi-omics, *SOCS3*

## Abstract

Meat quality is an important economic trait that influences the development of the pig industry. Skeletal muscle development and glycolytic potential (GP) are two crucial aspects that significantly impact meat quality. It has been reported that abnormal skeletal muscle development and high glycogen content results in low meat quality. However, the genetic mechanisms underlying these factors are still unclear. Compared with intensive pig breeds, Chinese indigenous pig breeds, such as the Jinhua pig, express superior meat quality characteristics. The differences in the meat quality traits between Jinhua and intensive pig breeds make them suitable for uncovering the genetic mechanisms that regulate meat quality traits. In this study, the Jinhua pig breed and five intensive pig breeds, including Duroc, Landrace, Yorkshire, Berkshire, and Pietrain pig breeds, were selected as experimental materials. First, the *F_ST_* and XP-EHH methods were used to screen the selective signatures on the genome in the Jinhua population. Then, combined with RNA-Seq data, the study further confirmed that *SOCS3* could be a key candidate gene that influences meat quality by mediating myoblast proliferation and glycometabolism because of the down-regulated expression of *SOCS3* in Jinhua pigs compared with Landrace pigs. Finally, through *SOCS3* knockout (KO) and overexpression (OE) experiments in mouse C2C12 cells, the results showed that *SOCS3* regulated the cell proliferation of myoblasts. Moreover, *SOCS3* is involved in regulating glucose uptake by the IRS1/PI3K/AKT signaling pathway. Overall, these findings provide a basis for the genetic improvement of meat quality traits in the pig industry.

## 1. Introduction

Long-term natural and artificial selection have made Chinese indigenous pigs famous all over the world for their characteristics, e.g., excellent meat quality, high reproductive ability, and strong disease resistance [1,2]. It is known that China has a vast territory, and diverse geographical environments have resulted in pigs with distinct characteristics in different regions. For example, Tibetan pigs are known for their strong adaptability, particularly to hypoxia environments [3]. Taihu pigs are renowned for their high-fertility characteristics [1,4]. Jinhua pigs are widely praised for their superior meat quality, and Jinhua ham is a well-known trademark [5]. Meat quality is an important economic trait that influences people’s consumption and desire for pork [6]. With the development of society, people’s requirements for meat quality are increasing. Therefore, it is meaningful to explore the genetic mechanisms underlying meat quality traits for the further development of the pig industry.

Meat quality traits include intramuscular fat (IMF), pH, water-holding capacity, meat color and tenderness, and so on [7]. Previous studies have shown that most meat-quality traits exhibit low-to-moderate heritability [8]. Historically, swine breeders have focused on growth performance but have overlooked meat quality [9,10]. Due to its genetic relationships with other economic traits, meat quality has become one of the primary objectives of pig breeding programs today [11,12,13]. However, the current understanding of Chinese indigenous pigs is limited to their phenotypes. Therefore, further exploration of meat quality-related genes is necessary owing to the insufficient research on the gene localization of meat quality traits.

In livestock industries, both skeletal muscle development and the glycolysis potential (GP) of skeletal muscle are two vital aspects that influence meat quality. GP is particularly used as a biomarker to predict meat quality for farm workers. Cell proliferation, cell cycle, cell apoptosis, and cell differentiation are four manifestations of skeletal muscle development. Additionally, the glycogen contents in skeletal muscle are the determinant of GP values. Ma et al. [14] reported that a 32-bp deletion in the open reading frame of *PHKG1* caused a high glycogen content and low water-holding capacity in pork. Additionally, *PRKAG3* has been elaboratively evidenced to impact glycogen and GP levels in Hampshire and its related synthetic lines, thus influencing meat quality [15,16]. In view of glucose uptake as a determinant that influences glycogen synthesis capacity [17], the research progress on the regulation mechanism of glucose uptake in skeletal muscle is still slow, and lots of unknowns are still worth exploring.

At present, it is convenient to obtain the genomic and transcriptional information of a population based on high-throughput sequencing techniques. In this study, the Jinhua pig and five intensive pig breeds (Duroc, Landrace, Yorkshire, Pietrain, and Berkshire) were chosen as the research objects. The differences in meat quality between Jinhua pigs and Western pigs make it easier to reveal the genetic mechanism of meat quality. Meat quality traits were formed through long-term selections so that the associated selective signatures could be detected, which might provide efficient clues regarding the molecular mechanisms of meat quality. Both the population differentiation coefficient (*F_ST_*) and cross-population extended haplotype homozygosity (XP-EHH) methods were identified as suitable methods for detecting selective signatures across populations [18,19]. Additionally, RNA-Seq data can comprehensively reflect transcription levels in the body. Therefore, with the aim of understanding the genetic mechanisms regulating meat quality, this study combined genome and transcriptome analysis to explore the candidate causal genes that might mediate meat quality through two different molecular levels, which were verified in vitro.

## 2. Results

### 2.1. Genetic Population Structure Analysis in Jinhua Pigs

This study was based on 8,528,947 SNPs uniformly distributed across the pig genome (Figure 1). NJ-tree, PCA, and population structure analysis were performed to test the consistency of the Jinhua pig population. Of particular interest is that Jinhua pigs and intensive pig breeds were separated into two clades by the NJ-tree and PCA. In the NJ-tree plot (Figure 2A), Jinhua pigs were clustered into one branch, and others, including Duroc, Landrace, Yorkshire, Pietrain, and Berkshire pigs, were clustered into another branch. The PCA plot showed that the contributions of the two PCs were 62.35% and 21.89%, respectively (Figure 2B). Herein, PC1 separated individuals into two groups: the Jinhua breed and the intensive breeds. Furthermore, population structure analysis showed two distinct subpopulations: Jinhua pigs and western breeds with K = 2. When K = 6, the cross-validation error, was the smallest value, and each pig breed could be distinguished clearly from others (Figure 2C). The overall results indicated the Jinhua pig population has good uniformity.

### 2.2. Screening for Candidate Genes Based on Selection Signature and RNA-Seq

To explore the genomic variability of meat quality in different pig breeds, the genomic selection signatures of Jinhua pigs, compared with five Western pig breeds, were screened at a population level. Two different approaches, *F_ST_* and XP-EHH, were applied to exhibit the distribution of SNPs with *F_ST_* or XP-EHH values across the genome (Figure 3A,B). The top 0.5% SNPs based on *F_ST_* values were considered as the candidate loci and 2757 genes were annotated (Appendix A). The SNPs based on XP-EHH values with *p* < 0.01 were considered the candidate loci, and a total of 488 genes were annotated (Appendix A). The RNA-Seq analysis of the longissimus dorsi muscle (LDM) showed 1508 differentially expressed genes (DEGs) between Jinhua and Landrace pigs. Compared with Landrace pigs, 803 genes were up-regulated, and 705 were down-regulated in Jinhua pigs (Figure 3C and Appendix A). To reduce the false positive results, the genes detected by at least two methods were selected as the candidates. According to Venn’s result, 266 candidate genes overlapped between the two groups (Figure 3D).

### 2.3. Functional Analysis of Candidate Genes

To explore the function of candidate genes, KEGG enrichment analysis was performed (*p* < 0.05). Genes annotated by *F_ST_*, XP-EHH, and RNA-Seq methods, respectively, were enriched in the corresponding KEGG pathways (Figure 4A–C, Appendix A). There were 29 common genes enriched in 12 common pathways in relation to energy metabolism, including the PI3K-AKT signaling pathway, MAPK signaling pathway, insulin signaling pathway, etc. (Figure 4D). Moreover, the protein–protein interaction network (PPI) analysis of these 29 common genes showed that *SOCS3*, *PRKCD*, *PAK1*, and *SORBS1* were associated with insulin sensitivity, which could be related to glucose uptake (Figure 4E). Additionally, *SOCS3* was the gene with the most significant difference in the expression level among these four genes above and between Jinhua pigs and Landrace pigs (Appendix A). Therefore, the functions of *SOCS3* in regulated myoblasts proliferation and glucose uptake need to be further explored. 

### 2.4. Effects of Socs3 on the Proliferation Ability of Myoblasts

To explore the role of *Socs3* for skeletal muscle, *Socs3* knockout (KO) and overexpression (OE) mouse C2C12 cells were constructed. qPCR analysis showed that the *Socs3* mRNA level significantly decreased and increased in *Socs3* KO and OE mouse C2C12 cells, respectively (Figure 5A,D). Interestingly, proliferation-related genes, including *Pcna*, *Ccnb1*, and *Ccnd1*, were down-regulated in *Socs3* KO cells (Figure 5A). The CCK8 and EdU cell proliferation assays further confirmed a decrease in the proliferation ability of *Socs3* KO cells (Figure 5B,C). Conversely, *Pcna*, *Ccnb1*, and *Ccnd1* were up-regulated in *Socs3* OE cells. CCK8 and EdU cell proliferation assays showed an increase in the proliferation ability of *Socs3* OE cells (Figure 5E,F). Then, the flow cytometry assay detected the cell cycle of *Socs3* KO and OE cells. The results found that *Socs3* KO decreased the percentage of cells in the G1 phase and increased cells in the G2 phase significantly (Figure 6A,B). Additionally, *Socs3* OE increased the percentage of cells in the G1 phase and reduced cells in the S and G2 phases significantly (Figure 6C,D). These results indicate how *Socs3* regulated the cell proliferation of myoblasts.

### 2.5. Socs3 Is Necessary for Glucose Uptake of Myotubes Cells

To identify the function of *Socs3* in regulating glucose uptake in skeletal muscles, *Socs3* KO and OE mouse C2C12 cells were induced with differentiation for 5 days and then collected for the experiments. The results showed that the *Socs3* mRNA level significantly decreased and increased in *Socs3* KO and OE C2C12 cells, respectively (Figure 7A,D). It is worth noticing that glucose-transfer-related genes, including *Irs1* and *Pcg-1α* mRNA levels and GLUT4, PI3K, and *p*-AKT protein levels, were obviously reduced in *Socs3* KO myotubes cells (Figure 7A–C). Additionally, *Irs1* and *Pcg-1α* mRNA, as well as the GLUT4, PI3K, and *p*-AKT protein levels, evidently increased in *Socs3* OE myotubes cells (Figure 7D–F). These results suggest that *Socs3* mediated the glucose uptake of myotube cells.

## 3. Discussion

Jinhua pigs, one of the Chinese indigenous pig breeds, are well-known for their superior meat quality [20]. Intensive pig breeds generally exhibit lower meat quality traits than Jinhua pigs. Therefore, exploring the differences between the Jinhua pig breed and intensive pig breeds could contribute to uncovering the genetic mechanisms that regulate meat quality, which is meaningful for the development of the pig breeding industry. Considering the introduction of intensive pig breeds, the number of Chinese indigenous pigs was decreasing yearly, which has led to the crossbreeding of Chinese local pigs. To test the consistency of these samples, three population structure analysis methods (NJ-tree, PCA, Structure) were used to detect the authenticity and reliability of the Jinhua pig population. These results revealed the Jinhua pig population was significantly distinguished from intensive pig populations, which reflected that Jinhua pigs had a pure pedigree in this study. Meanwhile, these results provided a reliable basis for subsequent research.

With the development of high-throughput sequencing technology, a single level of molecules did not meet the research requirements at all, and multi-omics analysis came into being. In this context, for studying meat quality traits holistically, genomics and transcriptomics data were analyzed and combinedly to identify the candidate genes. The selection signatures method is commonly used for genomic analysis and has observed great progress in pigs [1,4], chickens [21,22], cattle [23,24], etc. Both the *F_ST_* and XP-EHH methods are widely used for exploring selection signatures between populations [25]. In addition, RNA-Seq technology is generally used to identify the DEGs between two groups. Therefore, it might ensure the accuracy and reliability of the results and reduce false positive results through integrations of the *F_ST_*, XP-EHH, and RNA-Seq methods. We know that multi-breed comparisons can better reflect the genetic differences in meat quality between Jinhua pigs and Western pigs. In future studies, we aim to continue to expand the data detection of different varieties.

There are many factors that influence meat quality, and skeletal muscle development is one of them. The rates of cell proliferation and cell cycle could reflect the conditions of skeletal muscle development [26]. Until now, the roles of *SOCS3* in disease treatment and immune response have not been identified [27,28]. It remains unclear whether *SOCS3* is involved in regulating skeletal muscle development. Therefore, we performed cell proliferation and the cell cycle of the myoblast in the current study. After the successful KO of *Socs3*, C2C12 cells showed a significant decrease in cell proliferation, as well as the cell ratios of the G1 phase, and *Socs3* OE significantly increased cell proliferation, and the cell ratios in the G1 phase increased as well. These studies have confirmed that *Socs3* regulates cell proliferation and the cell cycle to achieve the regulation of skeletal muscle development.

The glycogen content in skeletal muscle is another important factor that influences meat quality, and high glycogen content can result in low meat quality [14,16]. Considering that glucose is the basic building block of glycogen, it follows that the glucose uptake capacity of skeletal muscle determines the glycogen synthesis capacity [29,30]. Thus, it is necessary to explain the genetic mechanism in regulating the glucose uptake capacity of the skeletal muscle. Previous studies have reported that *SOCS3* is a target for treating metabolic disorders and influences insulin sensitivity and glucose homeostasis in the body [31]. The insulin signaling pathway has a stimulatory effect on glucose uptake in skeletal muscle [32,33]. Additionally, the functions of *SOCS3* in regulating energy metabolism and glycometabolism in tissues, such as the liver and adipose, have been reported widely [34,35]. However, the underlying mechanism of *SOCS3* in the glucose uptake of skeletal muscle remains unclear as yet. In this study, by integrations of multi-omics analysis, *SOCS3* was enriched significantly in Jinhua pigs, and a lower expression of *SOCS3* was observed in Jinhua pigs compared to Landrace pigs. Therefore, it is necessary to verify the function of *SOCS3* in regulating glucose uptake in skeletal muscle.

GLUT4 is the major glucose transport protein of skeletal muscle, which determines the glucose uptake level. Additionally, the PI3K-AKT signaling pathway is a classic pathway in muscle glucose metabolism. Skeletal muscle is the largest insulin receptor organ in the body when insulin binds to an insulin receptor, and activating IRS1 promotes the glucose uptake of skeletal muscle. Additionally, there was a paper published which suggested that *IRS1* could be regulated by *PCG-1α* [36]. Furthermore, Guo et al. [37] found that the GLUT4 expression was mainly influenced by the IRS1/PI3K/AKT signal pathway. PI3K interacts with p-IRS with subunit p85 to activate/phosphorylate its downstream effector AKT; this response results in the translocation of GLUT4 and then the acceleration of glucose uptake activity in myoblasts and adipocytes [38,39]. Therefore, we evaluated the expression of IRS1, PI3K, and AKT since they are major upstream regulators of GLUT4 translocation. Here, the results describe how *Socs3* KO myoblasts decreased the proteins of GLUT4, PI3K, and p-AKT, as well as the mRNA of *Irs1* and *Pcg-1α*, which is consistent with the observations reported previously. In addition, *Sosc3* OE myoblasts increased the proteins of GLUT4, PI3K, and p-AKT, and the mRNA of *Irs1* and *Pcg-1α* increased as well. These results imply that *Socs3* regulated the glucose uptake of myoblasts through the IRS1/PI3K/AKT signal pathway. Additionally, these results also suggest the reason why Jinhua pigs have better meat quality than intensive pigs may due to their lower expression of *SOCS3*, which reduces glucose uptake and results in decreased glycogen synthesis.

## 4. Materials and Methods

### 4.1. Genotype and Quality Control

A total of 558 pigs from six breeds were selected (Appendix A). The DNA of Jinhua pigs was carried out from blood samples and genotyped using the GeneSeek GGP-Porcine chip (Neogen Corporation, Lansing, MI, USA). The SNP data of Duroc, Landrace, Large White, Berkshire, and Pietrain were obtained from the paper published by Yang et al. [40]. Genotype imputation was performed using Beagle version 5.0 via PHARP [41,42]. The quality control of SNPs was implemented using VCFtools version 0.1.14 [43], following the below criteria: (1) retaining the SNPs with a DR2 ≥ 0.5; (2) removing the SNPs with a call rate < 0.9 or MAF < 0.01; (3) retaining sites with a mean depth ≥ 3. Finally, the data set contained 8,528,947 autosome SNPs for further analysis.

### 4.2. Population Structure

To verify the consistency of the Jinhua pig population, we put Jinhua pigs together with other western pigs to make a population structure analysis. A molecular phylogenetic tree was constructed by the neighbor-joining (NJ) method in MEGA X, and the NJ-tree was visualized using FigTree v1.4.4. Principal component analysis (PCA) was conducted using Plink v1.9. software, and “PC1”, and “PC2” were used to distinguish the population structure. The result of PCA was visualized by the ggplot2 package. The population structure analysis was used by ADMIXTURE36 v1.3.0 [44]. The number of ancestral clusters (K) was set from 2 to 6, and a six-fold cross-validation was run to determine the K value with the lowest cross-validation error.

### 4.3. Selective Signatures Analysis and RNA-Seq Analysis

Two methods for genomic selection signature detection, including *F_ST_* and XP-EHH, were performed to detect the genomic regions under selection in Jinhua pigs by comparing them with five western pig breeds. The *F_ST_* value of each SNP between the Jinhua pigs and other pigs was calculated with vcftools v0.1.13. The top 0.05% of SNPs were empirically selected as potential candidate loci under a positive selection. The XP-EHH statistic was designed to detect ongoing or nearly fixed selection signatures by comparing the haplotypes from two populations [45]. XP-EHH values were estimated using Selscan v1.3.0, and the XP-EHH value of each SNP was normalized [46]. Extremely high values in the *p* < 0.005 right-tail of XP-EHH were selected as potential candidate loci under positive selection as well. The raw data of RNA-Seq of LDM between Jinhua and Landrace pigs were downloaded from the dataset GEO: GSE113237.

### 4.4. Functional Annotation

To define the actual function of the candidate genes above, the Kyoto Encyclopedia of Gene and Genomes (KEGG) pathway enrichment analysis was performed using KOBAS v3.0 [47]. The *p* < 0.05 of KEGG pathways were regarded as significant results.

### 4.5. Cell Culture and Treatment

C2C12 and 293T cell lines were used in this study and purchased from the National Collection of Authenticated Cell Cultures (Shanghai, China). The cells were cultured in a growth medium (GM) consisting of DMEM (Gibco, Grand Island, NY, USA) plus 10% fetal bovine serum (FBS; ExCell, Shanghai, China) and 1% penicillin-streptomycin (Gibco, Grand Island, NJ, USA) in a 5% CO2 humidified incubator at 37 °C. After C2C12 cells reached 80–90% confluence, the differentiation medium (DM), supplemented with 2% horse serum (HS; Gibco, USA), replacing FBS, was used to induce myogenic differentiation.

### 4.6. SOCS3 Knockdown and Overexpression

For lentiviral infection experiments, plasmid lentiCRISPRv2, psPAX2, and pMD2.G were necessary. At first, sgRNA (5′-TACTGGAGCGCCGTGACCGG-3′) was designed through an online website (https://crispr.dbcls.jp, accessed on 7 November 2020), which was subcloned into the lentiCRISPRv2 vector between BglII/HpaI restriction enzyme sites. The recombined lentiCRISPRv2 vector was co-transfected into 293T cells with psPAX2 and pMD2.G plasmids using the Lipofectamine 2000 reagent (Invitrogen, Waltham, MA, USA). After 48 h and 72 h, the supernatants were harvested and filtered through 0.22 μm pore filters (Sartorius, Göttingen, Germany) to gain lentiviruses. Finally, C2C12 myoblasts were infected with lentiviruses plus 4 μg/mL polybrene (Beyotime Biotechnology, Shanghai, China) and sieved with 3 μg/mL puromycin (Gibco, USA). *SOCS3* overexpression plasmids were purchased from Hanbio (Shanghai, China).

### 4.7. Cell Proliferation and Cell Cycle Assay

For the cell proliferation assay, the transfected myoblasts were cultured in a 12-well plate detected by the EdU assay kit (Vazyme, Nanjing, China) and following the manufacturer’s protocol. Images were obtained by fluorescence microscopy (Nikon, Tokyo, Japan). Cell proliferation was also treated with a CCK-8 assay kit (Solarbio, Beijing, China). The transfected myoblasts were cultured in a 96-well plate treated with 10 μL of CCK-8 after incubation at 37 °C for 2 h and the absorbance was detected at 450 nm using a microplate reader (BioTek, Winooski, VT, USA). To detect the cell cycle, the transfected myoblasts were cultured in a 6-well plate and stained with propidium iodide. Then, the ratios of the cells in the G1, S, and G2 phases were counted and compared using flow cytometry (BD Biosciences, Franklin Lakes, NJ, USA).

### 4.8. Quantitative Real-Time PCR

The total RNA was extracted from C2C12 using the TRIzol reagent (Thermo, Waltham, MA, USA). mRNA, which was reverse-transcribed into cDNA using the reverse transcription kit (Vazyme, China), and cDNA was analyzed by an SYBR Green PCR mix kit (Vazyme, China) in the QIAquant96 2plex system (Qiagen, Hilden, Germany). Relative gene expression was calculated using the 2^−ΔΔCt^ method and normalized to *Actb*. The primers are shown in Appendix A.

### 4.9. Western Blot

The total proteins were isolated from C2C12 using RIPA lysis buffer plus protease and phosphatase inhibitors (Beyotime Biotechnology, China). The proteins were boiled with a 5× SDS loading buffer for 10 min at 100 °C for denaturation. The proteins were separated by SDS-PAGE and transferred to PVDF membranes (Millipore, Burlington, MA, USA). The membranes were blocked with 5% non-fat milk at room temperature for 1 h and were subsequently incubated with primary antibodies overnight at 4 °C. The next day, the membranes were washed three times using 1× TBST and were then incubated with secondary antibodies at room temperature for 1 h. At last, the membranes were washed three times and exposed to an ECL Reagent (Beyotime Biotechnology, China). The WB results were photographed by the SHST Capture system (Shenhua, Hangzhou, China) and quantified using ImageJ v.1.8.0. The antibodies are shown in Appendix A.

### 4.10. Statistical Analysis

In this study, the statistical significance of differences was determined using an unpaired Student’s *t*-test with GraphPad Prism 8. The data were presented as the mean ± SD, * represents *p* < 0.05, ** represents *p* < 0.01, and *** represents *p* < 0.001.

## 5. Conclusions

Taken together, *SOCS3* was simultaneously annotated through combination analysis based on genomics and transcriptomics and regarded as the candidate gene that could regulate the meat quality of pigs. The results indicate that *SOCS3* not only influences myoblast proliferation and cell cycle but also regulates the glucose uptake of myoblasts through the IRS1/PI3K/AKT signaling pathway in mouse C2C12 cells. These findings provide a theoretical basis for further insights into the regulating mechanism of meat quality traits in pigs. In addition, we still need to further explore the function of *SOCS3* in regulating porcine skeletal muscle cells.

## Figures and Tables

**Figure 1 ijms-24-10593-f001:**
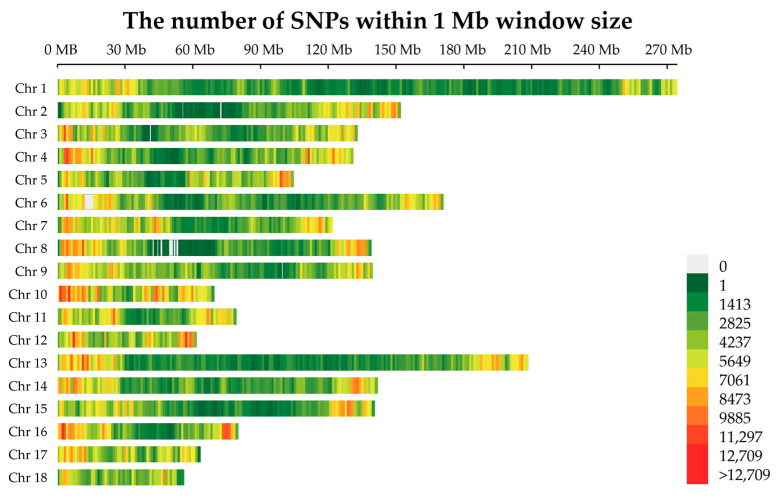
Distribution of SNPs on chromosomes. The x-axis represents the chromosomal position, and the y-axis represents chromosomes. Different colors represent SNP density in each 1 Mb genome block.

**Figure 2 ijms-24-10593-f002:**
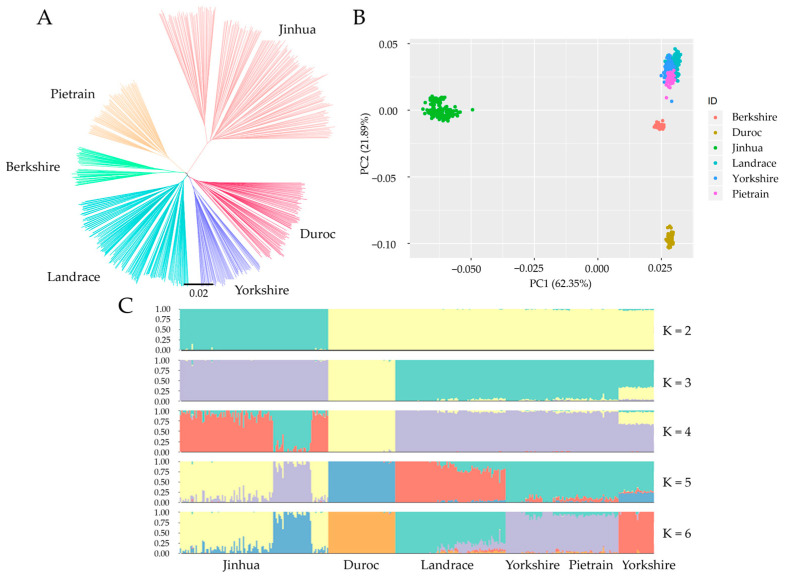
Population structure analysis between Jinhua, Duroc, Landrace, Yorkshire, Pietrain, and Berkshire pig breeds. (**A**) The NJ-tree analysis of all individuals. (**B**) The PCA analysis of all individuals. The x-axis denotes the first principal component, and the y-axis represents the second principal component. (**C**) The population structure of all individuals. K represents the number of possible ancestral varieties.

**Figure 3 ijms-24-10593-f003:**
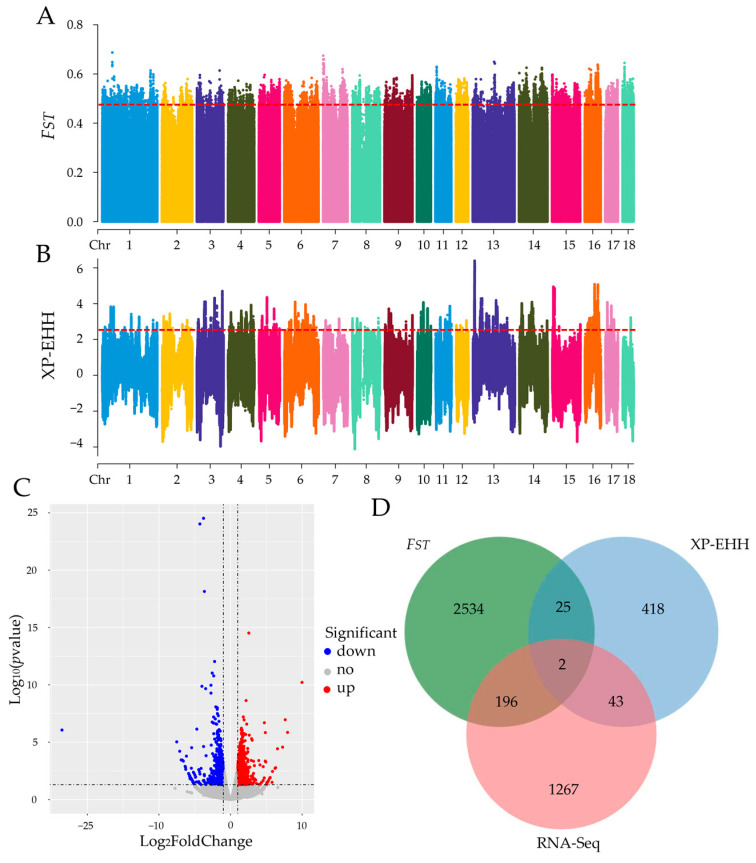
Selection signature analysis and RNA-Seq analysis. (**A**) The distribution of selection signatures between Jinhua pigs and Western pigs was detected by the *F_ST_* method. The x-axis represents the chromosome location of SNPs; the y-axis represents the *F_ST_* values of SNPs. (**B**) The distribution of selection signatures between Jinhua pigs and western pigs was detected by the XP-EHH method. The x-axis represents the chromosome location of SNPs, and the y-axis represents the XP-EHH values of SNPs. (**C**) The volcano plot of DEGs based on RNA-Seq analysis. The red dots, blue dots, and grey dots represent the up-regulated genes, down-regulated genes, and no differentially expressed genes in Jinhua pigs compared with Landrace pigs, respectively. (**D**) The Venn analysis of different genes annotated by *F_ST_*, XP-EHH, and RNA-Seq methods.

**Figure 4 ijms-24-10593-f004:**
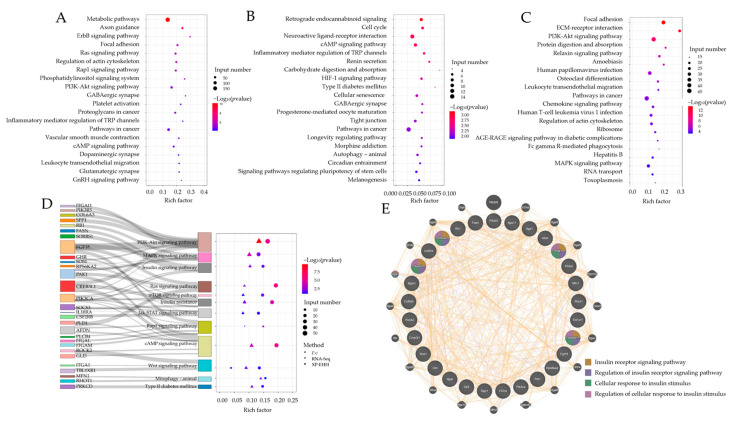
The function analysis of candidate genes. (**A**) The KEGG enrichment of genes selected by the *F_ST_* method. (**B**) The KEGG enrichment of genes selected by the XP-EHH method. (**C**) The KEGG enrichment of genes annotated by the RNA-Seq method. (**D**) The KEGG enrichment analysis of common candidate genes found by *F_ST_*, XP-EHH, and RNA-Seq methods. (**E**) The PPI interaction network analysis of candidate genes.

**Figure 5 ijms-24-10593-f005:**
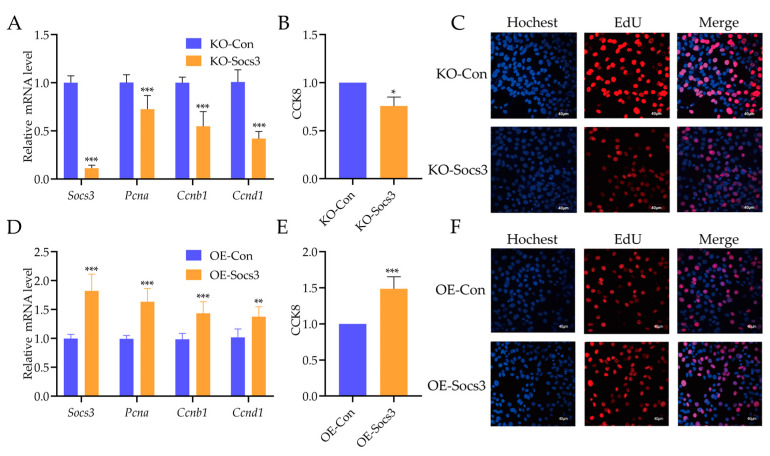
*Socs3* influences the proliferation of myoblasts. (**A**) The mRNA levels of *Socs3*, *Pcna*, *Ccnb1*, and *Ccnd1* in *Socs3* knockdown C2C12 cells using qPCR. (**B**) The CCK-8 assay was used to measure the proliferation of *Socs3* knockdown C2C12 cells. (**C**) The EdU assay was used to measure the proliferation of *Socs3* knockdown C2C12 cells. (**D**) The mRNA levels of *Socs3*, *Pcna*, *Ccnb1*, and *Ccnd1* in *Socs3* overexpression C2C12 cells using qPCR. (**E**) The CCK-8 assay was used to measure the proliferation of *Socs3* overexpression C2C12 cells. (**F**) The EdU assay was used to measure the proliferation of *Socs3* overexpression C2C12 cells. The scale bar indicates 40 μm. *Actb* was used as a loading control in qPCR. *: *p* < 0.05, **: *p* < 0.01, ***: *p* < 0.001.

**Figure 6 ijms-24-10593-f006:**
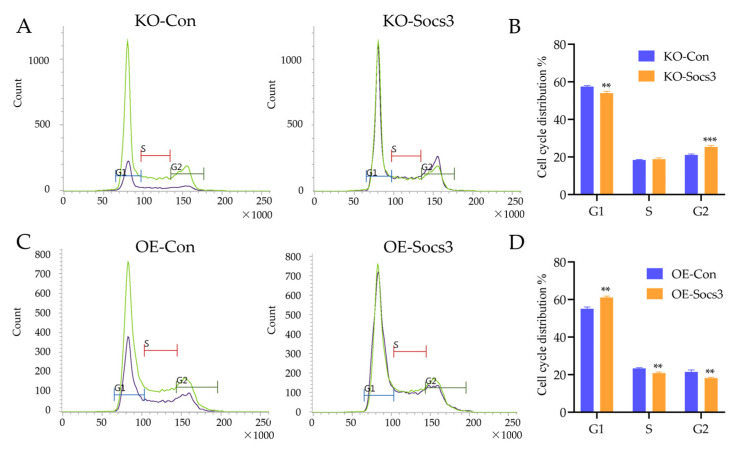
*Socs3* influences the cell cycle of myoblasts. (**A**) The cell cycle distribution of *Socs3* knockdown C2C12 cells was evaluated using a flow cytometry assay. (**B**) Statistical analysis of the percentage of *Socs3* knockdown C2C12 cells in the G1, S, and G2 phases. (**C**) The cell cycle distribution of *Socs3* overexpression C2C12 cells was evaluated using a flow cytometry assay. (**D**) The statistical analysis of the percentage of *Socs3* overexpression C2C12 cells in the G1, S, and G2 phases. **: *p* < 0.01, ***: *p* < 0.001.

**Figure 7 ijms-24-10593-f007:**
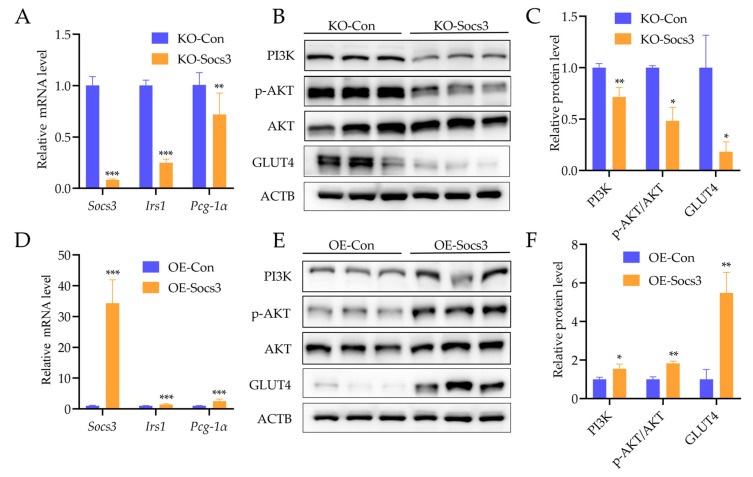
*Socs3* mediates the glycogen uptake of myotubes. (**A**) mRNA expression levels of *Socs3*, *Irs1* and *Pcg-1α* in *Socs3* KO C2C12 cells by qPCR. (**B**,**C**) The protein expression levels of GLUT4, PI3K, AKT and p-AKT in *Socs3* KO C2C12 cells by Western blot and quantification analysis. (**D**) mRNA expression levels of *Socs3*, *Irs1,* and *Pcg-1α* in *Socs3* OE C2C12 cells by qPCR. (**E**,**F**) The protein expression levels of GLUT4, PI3K, AKT, and p-AKT in *Socs3* OE C2C12 cells by Western blot and quantification analysis. Actb and ACTB were used as the loading controls in qPCR and Western blot respectively. *: *p* < 0.05, **: *p* < 0.01, ***: *p* < 0.001.

## Data Availability

The SNP chip datasets: 558 samples are available from the APLHADB (http://alphaindex.zju.edu.cn/ALPHADB/download.html, accessed on 23 June 2023). The RNA-Seq data of LDM between Jinhua and Landrace pigs from previous study is available from the NCBI (Bioproject number PRJNA450530).

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
