# Peer review of "The Role of SOCS3 in Regulating Meat Quality in Jinhua Pigs"

_ijms, 2023, doi:10.3390/ijms241310593_

Round 1
Reviewer 1 Report
I’ve carefully read this interesting manuscript by Wu and colleagues. The study they presented is well conceived and performed, and it adds, in my opinion, some novel and useful knowledge in its field. The study possesses a very complete and well-structured design, which allows a deep investigation of the topic and the obtention of very interesting and novel findings. The manuscript deserves to be published, anyway, some parts should be carefully rewritten since in its current form some major flaws impair the fully understanding and exploitation of these findings.
Here below the major points that should be amended:
- The discussion needs to be almost completely reorganized. The authors mainly focused this section on discussing the methodological aspects of their work, relegating the discussion on biological aspects to the last paragraphs. I know that the technical work is very well done and that it deserves to be put in light, anyway, also the findings should be deeply explained from a biological point of view. I can give you two examples: the authors should explain, or propose, the potential link(s) between glucose metabolism and meat quality, and also try to better place SOC3 role in this link. Second, the authors stated the SOC3 enhances myoblasts proliferation, it would be interesting if the authors suggest how this biological effect could improve meat quality.
- The authors include the introduction with a paragraph in which they anticipated the results. I think that this paragraph should be replaces by a clear presentation of the study design. As mentioned here above, the study is well developed and complex, a clear rationale with the chronological presentation of the approach would help the reader in understanding the results.
Some other minor points are reported here below:
- The manuscript would take from of a thorough revision by a native English speaker.
- Some statements are missing of bibliographic citation (1st sentence of the introduction, 1st sentence of the 2nd paragraph of the introduction, 1st sentence of the discussion).
- Page 3: why did you present, in the text, only the comparison with Landrace omitting the other breeds?
the English needs a thorough revision.
Author Response
- The discussion needs to be almost completely reorganized. The authors mainly focused this section on discussing the methodological aspects of their work, relegating the discussion on biological aspects to the last paragraphs. I know that the technical work is very well done and that it deserves to be put in light, anyway, also the findings should be deeply explained from a biological point of view. I can give you two examples: the authors should explain, or propose, the potential link(s) between glucose metabolism and meat quality, and also try to better place SOC3 role in this link. Second, the authors stated the SOC3 enhances myoblasts proliferation, it would be interesting if the authors suggest how this biological effect could improve meat quality.
Authors: Thanks. The discussion has been almost completely reorganized. The last three paragraphs of the discussion are deeply explained from a biological point of view. We explain the potential link between glucose uptake capacity and meat quality, and also better place SOC3 role in this link. Additionally, we explain myoblasts proliferation is an important aspect of skeletal muscle development which influences meat quality. Please see the discussion section.
- The authors include the introduction with a paragraph in which they anticipated the results. I think that this paragraph should be replaces by a clear presentation of the study design. As mentioned here above, the study is well developed and complex, a clear rationale with the chronological presentation of the approach would help the reader in understanding the results.
Authors: Thanks. A paragraph within the introduction which anticipates the results has been reorganized. We provide a clear rationale with the chronological presentation of the approach to help the reader in understanding the results. Please see the introduction section.
- The manuscript would take from of a thorough revision by a native English speaker.
Authors: Thanks. This manuscript has been thoroughly edited by a colleague fluent in English writing.
- Some statements are missing of bibliographic citation (1st sentence of the introduction, 1st sentence of the 2nd paragraph of the introduction, 1st sentence of the discussion).
Authors: Thanks. We have added the bibliographic citations of 1st sentence of the introduction, 1st sentence of the 2nd paragraph of the introduction, and 1st sentence of the discussion. Please see line 32, line 44, and line 208.
- Page 3: why did you present, in the text, only the comparison with Landrace omitting the other breeds?
Authors: Thanks. At genome level, Jinhua pig breed and five intensive pig breeds (Duroc, Landrace, Yorkshire, Berkshire and Pietrain) were used for detecting selective signatures across the genome in Jinhua population by FST and XP-EHH methods. At transcriptome level, it found that most of the RNA-Seq data of Jinhua pig skeletal muscle were compared with Landrace pigs according to NCBI public data. Therefore, Landrace pigs were selected for comparison.
Reviewer 2 Report

There are so many minor spelling and grammar errors, please check the spelling and grammar carefully!
Author Response
- there are so many little typos and grammar errors in the manuscript. Please check the spelling and grammar carefully.
Authors: Thanks. The typos and grammar errors in the manuscript have been carefully corrected.
- Results 2.1. (Fig.1A). There are no 1A, 1B.
Authors: Thanks. The note of Fig.1 has been corrected. please see line 88.
3、Page 4, “And SOCS3 was the largest DEG between Jinhua pigs and Landrace pigs (Fig. 4E, Table S10). There- fore, we further explore the function of SOCS3.” It is very difficult to read the gene names in Fig 4E (due to the quality of the original figure or the file for review having low quality?), and I can not find Table S10.
Authors: Thanks. We have re-uploaded Fig.4E. Table S10 is in the attached excel file.
- I like the SOCS3 knockout and overexpression experiment, which experimentally validated SOCS3 function. I also noted that the experiments were done in two different cell lines. One is a mouse cell line, and one is a human cell line. Was any consideration taken for the selection of cell lines?
Authors: Thanks. Mouse C2C12 cells are skeletal muscle cells, which can replace porcine skeletal muscle satellite cells to a certain extent to verify the regulatory effect of SOCS3 on skeletal muscle. Human 293T cells are common tool cells and were used for lentivirus packaging in this study. These two cell lines are not in conflict.
Reviewer 3 Report
Overall the manuscript is well written and the information is of significant interest to the Journal's readers. The topic of the manuscript is proper for the Journal.
I have some minor concerns:
Please, avoid the use of personal form (we, our, etc.) throghout the manuscript.
What about the inclusion/exclusion criteria for the enrollment of animals in the study?
What about the health status of animals at the enronllment?
Regarding the statistical analysis of data, did Authors apply a normality test in order to verify the normal distribution of data?
Figure are nice and well represent the findings of the study. However, please improve the quality/size of figures 1, 2, 3 and 4.
I suggest to rewrite the conclusion section in order to better summarize the findings ad amphasize the significance of the study.
Author Response
- Please, avoid the use of personal form (we, our, etc.) throghout the manuscript.
Authors: Thanks. The personal form (we, our, etc.) throughout the manuscript has been avoided.
- What about the inclusion/exclusion criteria for the enrollment of animals in the study?
Authors: Thanks. Through PCA and NJ-Tree analysis, individuals of the same breed are clustered together, and different breeds are separated from each other indicating the pure descent of all individuals, which can be used for analytical research.
- What about the health status of animals at the enronllment?
Authors: Thanks. All individuals in the study were in good health.
- Regarding the statistical analysis of data, did Authors apply a normality test in order to verify the normal distribution of data?
Authors: Thanks. We apply a normality test found that the XP-EHH values of SNPs were in normal distribution, and significance test was performed.
- Figure are nice and well represent the findings of the study. However, please improve the quality/size of figures 1, 2, 3 and 4.
Authors: Thanks. We have improved the quality/size of Fig. 1, Fig. 2, Fig. 3 and Fig. 4.
- I suggest to rewrite the conclusion section in order to better summarize the findings ad amphasize the significance of the study.
Authors: Thanks. We have rewrote the conclusion section and better summarize the findings ad emphasize the significance of the study. Please see conclusion section.
Round 2
Reviewer 1 Report
I want to thank the authors for their efforts in improving the text. The manuscript is now suitable for publication.
English needs just some minor revision during the editing process.